# Antimicrobial Stewardship Programs in Latin America and the Caribbean: A Story of Perseverance, Challenges, and Goals

**DOI:** 10.3390/antibiotics12081342

**Published:** 2023-08-21

**Authors:** Natalia Restrepo-Arbeláez, Juan Carlos Garcia-Betancur, Christian Jose Pallares, María Virginia Villegas

**Affiliations:** 1Grupo de Investigación en Resistencia Antimicrobiana y Epidemiología Hospitalaria (RAEH), Universidad El Bosque, Bogotá 110121, Colombia; nrestrepoa@unbosque.edu.co (N.R.-A.); betancurjuan@unbosque.edu.co (J.C.G.-B.); icako@hotmail.com (C.J.P.); 2Clínica Imbanaco Grupo Quirónsalud, Cali 760042, Colombia

**Keywords:** antimicrobial stewardship (ASM), Latin America (LATAM), Latin America and the Caribbean (LAC), antimicrobial resistance (AMR)

## Abstract

Antimicrobial resistance is one of the major global health threats. Antimicrobial stewardship (AMS) has been set as a priority within international action plans to combat this issue. The region of Latin America and the Caribbean are recognized for their high antimicrobial resistance rates; nevertheless, a low number of studies describing implemented interventions for this topic have been published. This review aims to provide an overview of the status of AMS in our region, focusing on the main progress achieved and describing the different published efforts made by countries towards the implementation of antimicrobial stewardship programs (ASP). Common areas of intervention included were (a) education approaches, (b) antimicrobial guideline implementation and monitoring, (c) diagnostic stewardship, (d) technological tools: electronic clinical decision support systems in AMS, (e) pharmacy-driven protocols and collaborative practice agreements, and (f) economic impact. The search demonstrated the varied interventions implemented in diverse healthcare settings; the results accentuate their influence on antimicrobial consumption, antimicrobial resistance, clinical outcomes, and direct economic impact. The integration of multiple strategies within each hospital was highlighted as an essential key to ASP success. Even though the literature found demonstrated clear progress, there is still a special need for strengthening leadership from the top down, defining goals based on needs, and gaining support through policy and financing in LAC.

## 1. Introduction

A global call to stop the spread of antimicrobial resistance (AMR) has been set as a priority by the World Health Organization (WHO) and other global health institutions [1,2,3,4]. AMR is recognized as a global major public health threat, with 1.27 million deaths yearly caused directly by it, and 4.95 million deaths associated with bacterial AMR [1]; it is also acknowledged that estimates will exponentially increase in the future [5]. The impact of AMR is not only related to mortality but also adds great burden of disease, years of life lost (YLL), and disability-adjusted life-years (DALYs) [1]. In addition, the economic impact of this issue is projected to be USD 100 trillion or even higher if actions to prevent it are not taken soon [2]. Thus, antimicrobial stewardship (AMS) has risen within national and international action plans to combat AMR [4]. Important advances have been made in AMS strategies and antimicrobial stewardship programs (ASP), particularly in high-income countries (HICs) [6,7,8]. Nevertheless, little data are available from low- and middle-income countries (LMIC) [2,9,10].

The region of Latin America and the Caribbean (LAC) has reached particularly high rates of AMR and extended dissemination of multidrug-resistant (MDR) pathogens, including carbapenem-resistant Gram-negative bacteria [10,11]. In this geographical region alone, during 2019, the deaths, YLL, and DALYs attributable to bacterial AMR were 14.4 per 100,000 (10.3–20.0), 405 per 100,000 (284.8–566.6), and 408 per 100,000 (286.9–570.0), respectively, compared to HIC, where deaths reached 13.0 per 100,000 (9.1–18.2), YLL reached 220.4 per 100,000 (149.9–314.0), and DALYs reached 222.3 (151.5–315.9) [1]. Nonetheless, publications in indexed databases addressing ASP in LAC are scarce [12].

The Centers for Disease Control and Prevention (CDC) and WHO have identified some key elements for AMS and the implementation of ASP, such as hospital leadership and formal support, accountability, the dedication of appropriate human resources such as pharmacy expertise, the implementation of evidence-based fundamental interventions that have demonstrated improved prescribing and patient care, tracking and monitoring progress, and sharing outcomes and success with the actors involved [12,13].

This review aims to provide an overview of the status of AMS in the LAC region, focusing on the main progress achieved during the last decade and describing the different published efforts made by countries in the region for the implementation of AMS strategies. The following common topics were found through our search and are recognized as fundamental interventions for AMS [13,14]; they are especially useful to identify ASP approaches, considering their adaptability within institutions, regardless of size and AMS personnel: (a) education approaches, (b) antimicrobial guideline implementation and monitoring, (c) diagnostic stewardship, (d) technological tools: electronic clinical decision support systems in AMS, and (e) pharmacy-driven protocols and collaborative practice agreements (CPA) [13,14]. Another interesting topic, which we titled economic impact, was observed in some of the publications and added to the review. This last topic is of utmost importance for healthcare administrators to support AMS as it shows its impact on decreasing unnecessary costs and improving advocacy for these strategies [15,16].

We searched data published from 2014 to April 2023 in the LAC region, including Argentina, Bolivia, Brazil, Chile, Colombia, Costa Rica, Ecuador, El Salvador, French Guiana, Guatemala, Honduras, Mexico, Nicaragua, Panama, Paraguay, Peru, Uruguay, Venezuela, the Greater Antilles (Cuba, Dominican Republic, Haiti, Jamaica, Puerto Rico, and the Lesser Antilles (Guadeloupe, Trinidad, and Tobago)). Initially, all documents (articles, reports, letters, editorials, comments, posters, theses, etc.) in English, Portuguese, or Spanish describing any AMS intervention (program, strategy, policy) aimed at optimizing the use of antimicrobials in healthcare settings of the countries listed above (LAC region) were considered. Additional studies were extracted from secondary sources, such as previously published reviews. Exclusion of publications was based on if the publication was not an official report or peer-reviewed article, if it only described resistance levels or disease management not related to AMS, if it did not take place in a healthcare setting, if it referred to medicine other than antimicrobials, or if it reported on AMR surveillance or economic impacts not related to an AMS intervention. Duplicates were deleted before screening. Titles and abstracts were screened by NRA and JCGB; afterwards, the publications were classified based on the categories previously described using Rayyan software [17]. Discrepancies were solved among all authors. Selected documents were reviewed for full-text eligibility. The quality of the studies was not formally evaluated. The search was conducted using specific databases: PubMed, Scopus, SciELO, Google Scholar, and LILACS using the terms ‘antimicrobial stewardship’ or ‘antimicrobial stewardship program’ together with an individual term for each of the countries mentioned above. All authors contributed to the analysis of each publication selected for Section 2. For the section’s expert comment and call for the future, all authors used existing and reviewed literature to express and discuss their analysis and draw conclusions on the status of AMS in LAC and what the region should focus on for AMS progress.

During the initial search, 65 publications were retrieved. Nine duplicates were deleted; fifty-six publications were screened to confirm geographical affiliation and abstract content using the selection criteria described above, excluding twelve. After full-text screening of the remaining forty-four publications, an additional six were excluded. A final number of thirty-eight published manuscripts were finally included in the review.

## 2. Status of Published Interventions of AMS in LAC

Multiple strategies must be deployed for ASP to make an impact on healthcare environments, and each setting creates a unique set of needs to improve AMS [14]. Some publications on this topic have shown the evolution of ASM in LAC and have highlighted the challenges to implementing ASP in the region (a lack of infrastructure, marked behavioral determinants, and limited resources) [10]; others have discussed ASP strategies deployed during the COVID-19 pandemic within LAC countries [18], and scoping reviews have also pointed out that there is mostly grey literature on how Latin American healthcare workers have implemented interventions, from education strategies to diagnostic technologies [12]. Despite this, in LAC, some publications have reviewed, described, and evaluated the impact of AMS strategies [9,10,12,19,20]. Not all of them describe the interventions and methods implemented for these strategies, making them hard to analyze. Based on the classification framework presented above, the following sections will describe the different AMS approaches used and implemented by healthcare institutions in LAC (Figure 1), highlighting the relevance of the results obtained.

### 2.1. Education Approaches

Education is a key component of AMS [13]. Publications from the LAC region in which educational-based interventions were highlighted demonstrated different approaches for implementation, including online strategies, in-person activities with healthcare personnel, periodic education meetings, and evaluations of impacts on clinical settings [21,22,23]. Díaz-Madriz et al. conducted a retrospective observational study in Costa Rica comparing prescription tendencies and AMR between two periods (2014–2015 and 2016–2017). Educational activities performed by their pharmacist-led AMS team promoted the use of first-generation cephalosporins as the drug of choice for surgical prophylaxis, instead of other antibiotics; at the end of the 2016–2017 period, there was a significant decrease in the use ceftriaxone of 19.1%, and a decrease in the use of levofloxacin of 49.9%, which authors think may have been explained by their intervention [21]. Another retrospective observational study conducted in four healthcare institutions in Colombia applied multiple strategies for AMS [22]. Part of the periodical interventions carried out included educational activities intended to remind physicians about the use of the antimicrobial guidelines already in place. These authors highlighted the importance of sharing antimicrobial guidelines and having discussions within and among hospitals and healthcare professionals in addition to educating physicians on the use of the guidelines for follow-up success. Positive results in decreasing AMR and antibiotic consumption were observed after the implementation of all the different strategies, including education [22]. A different approach in a tertiary teaching hospital in São Paulo, Brazil was the development of an innovative AMS intervention designed through a Moodle-based distance learning intervention consisting of a five-module course on the use of antimicrobial agents, and the prevention of antimicrobial resistance was evaluated over a 7-year period [23]. The course was offered to fifth-year medical students and aimed to determine if there was quantitative knowledge gained by participants. Pre- and post-intervention assessments were conducted, with a significant increase in the mean scores after the intervention (*p* < 0.001); 10.9% of students had a passing grade prior to the course; in contrast, 87.6% of students achieved a passing grade in the post-intervention scores. The authors proposed that to meet the ‘education’ needs for AMS and ASP, teaching hospitals should integrate the fundamental principles of antibiotic stewardship into their medical school curriculum [23]. Educational activities with training and practical exercises in the use of guidelines have also proven to be useful in educating healthcare personnel [22,24] and, sometimes, they are what contribute the most to increasing AMS development in hospitals [25].

### 2.2. Antimicrobial Guideline Implementation and Monitoring

Pallares et al. [23] published a prospective observational study in two intensive-care units (ICU) in Colombia. A detailed and tailored guideline for the use of antibiotics was created based on ICU microbiological surveillance and epidemiological data; it considered first-line antibiotics and alternative therapeutical options in case of treatment failure. In the second stage, these guidelines were distributed to ICU healthcare personnel, and training was conducted on how to utilize them. A new prescription format was implemented which contained useful information for infection definition and allowed for the reporting of treatment failure based on clinical and microbiological evaluation; the use of this format, which was then uploaded to the pharmacy’s software, helped to provide day-to-day monitoring in terms of treatment choice and guideline adherence. A 7-month pre-intervention and 7-month post-intervention period evaluation was performed. The results showed 83% adherence to the guidelines during the post-intervention period for ICU-1, and 86% in ICU-2. There was also a reduction in antibiotic consumption, especially carbapenems (−30% in ICU-1, −60% in ICU-2). Additionally, a decreased incidence of extended-spectrum beta-lactamase (ESBL) in *Escherichia coli* and *Klebsiella pneumoniae* infections was observed [24]. In a third-level hospital in Medellin, Colombia, three interventions were implemented; one of them was the creation of specific guidelines based on the most prevalent infections in the institution. In general terms, the results showed that adherence to guidelines was 82%, AMR in *Pseudomonas aeruginosa* decreased by 10%, and the consumption of meropenem, vancomycin, and colistin had a statistically significant decrease of 35%, 4%, and 28%, respectively [26].

An interesting prospective cohort study conducted in a hematologic ward in Brazil by Rosa et al. [27] evaluated the association between adherence to an ASP and mortality in hospitalized cancer patients with febrile neutropenia. The main variable studied was adherence to the institutional antimicrobial guidelines created and socialized by the ASP. The results showed a significant reduction in mortality during the 28-day follow-up period for patients treated according to ASP guidelines, with an adjusted relative risk reduction in 28-day mortality of 64% compared to cases that were non-adherent to guidelines [27]. On the other hand, in the Colombian retrospective observational study assessing the impact of ASP in four healthcare institutions previously mentioned [22]; besides the educational interventions described above, the ASP team updated and shared the institutional antimicrobial guidelines with different clinical specialists to reach a tailored consensus before its implementation. A prospective audit and feedback strategy was implemented to follow the antimicrobial guidelines; there was an important decrease in antibiotic consumption following this intervention (−29% institution A, −45% institution B, −28% institution C, −20% institution D). The authors highlighted the importance of sharing and discussing the AMS guidelines to improve adherence by healthcare professionals [22].

A quasi-experimental multicenter study implemented in Peru was identified [28]. The aim of this study was to evaluate the diagnostic phase of ASP and its early implementation in high-complexity hospitals. Using a modified index for ASP (ICATB = Indice Composite de bon usage des AnTiBiotiques), the authors performed a diagnosis of the status of the ASP in the different institutions using group interviews as well as documentary analysis. Based on the results of the interviews, they prioritized areas for ASP implementation. Through these focus groups, a consensus was reached for the empiric management of common infections as well as for the treatment of MDR pathogens. Through this consensus, formal ASP teams were established in healthcare institutions with the creation of antimicrobial guidelines and algorithms. The evaluation of the ICATAB index during the diagnostic phase and early implementation of ASP phase showed a significant increase (6.75 vs. 13.75), especially regarding the formal establishment of ASP teams, program construction, and the application of the guidelines during follow-ups [28].

Based on this review, restrictive (preauthorization) antimicrobial measures seem to be one of the most common types of intervention implemented in LAC. As an example, in Ecuador, Romo-Castillo et al. [29] performed a retrospective study of ASM strategies implemented by ASP consisting of preauthorization for the use of carbapenems. An interrupted time series analysis demonstrated a decrease in the use of imipenem (−3.97), ceftriaxone (−1.12), and piperacillin/tazobactam (−2.67), in contrast to the use of meropenem, which showed a slight increase of 0.66; these changes were expressed as defined daily doses (DDD)/100 bed days. The authors concluded that there is a need for more comprehensive ASPs, considering that restrictive measures alone have the potential to bring an increase in other unexpected effects [29]. A study carried out in a pediatric healthcare institution in Panamá compared antimicrobial consumption and antimicrobial therapy costs during pre-intervention (2007–2010) and post-intervention (2011–2017) periods. The intervention comprised of the restriction of a list of antimicrobials, requiring preauthorization and analysis of appropriateness from infectious disease specialists as well as a maximum amount of 10 days of therapy approved by the pharmacy. If a longer duration of therapy was necessary, further analysis was required by the ASP team. The results of this study, defined as the first AMS experience in the country, indicated significant changes in antibiotic consumption, with a decrease in vancomycin, gentamycin, meropenem, imipenem, and some other antibiotics, but there was an increase in amikacin, piperacillin/tazobactam, cefepime, and levofloxacin. [30]. Additional studies from Brazil and Colombia using ‘restrictive bundles’ for antimicrobials as part of their ASM strategies have also been published [26,31]. The Colombian experience using restrictive measures is part of a study carried out in Medellin, Colombia, as previously described. A restrictive policy was applied to a list of antibiotics, requiring authorization after the first dose for further dispensation by an infectious disease specialist. Even though the combined strategies had a positive impact on antimicrobial consumption and prescription patterns, the publication discussed how the preauthorization of antibiotics may increase the prescription of other antimicrobials and resistance to unrestricted antimicrobials [26]. In the case of the Brazilian experience, they conducted an interrupted time series analysis to understand the impact of a restricted bundle of 13 antimicrobials and antibiotic consumption implemented by a multidisciplinary ASP team, with 24 h available consultation. The results showed a non-significant decrease in the level of consumption of the bundle of restricted antibiotics; instead, there was an increase in the global use of antibiotics in the institution [31].

ASM interventions based on prospective audits and feedback allow for the optimization of antimicrobial prescriptions, identifying opportunities for better management [14]. One of the first studies found based on this strategy, conducted between 2003 and 2008, was a Brazilian quasi-experimental study conducted at a cardiology hospital assessing three stages (stage 1: before ASP, stage 2: infectious disease specialist in charge of ASP, stage 3: infectious disease physician and pharmacist working together on ASP) [32]. The ASM activities of the ASP were based on a prospective audit giving feedback to prescribers during stages 2 and 3. During stage 2, there was a significant reduction in the consumption of all antimicrobials as a global trend but also a specific increase in the consumption of piperacillin/tazobactam and fluoroquinolones. During stage 3, a reduction in fluoroquinolones, clindamycin, and ampicillin/sulbactam consumption was reported, but again, an increase in the use of cephalosporins was observed. The authors concluded that the most important gain was obtained during the antimicrobial policy implemented by multidisciplinary health professionals in the hospital [32]. In 2019, the Brazilian Health Maintenance Organization (HMO) implemented an outpatient parenteral antimicrobial therapy optimization program using prospective audits and feedback from an infectious disease specialist. The study retrospectively compared the intervention implemented during an 11-month period to data from the year prior to program initiation. During the intervention, the infectious disease physician reviewed the use of parenteral antimicrobials requested at the outpatient pharmacy for chronic patients followed at the outpatient clinic and patients discharged from hospitals by the HMO. A total of 506 requests were evaluated and, in general terms, audit interventions achieved important improvements in the correct change of antimicrobial (40%), reduction in treatment duration (25%), and change of administration route (24%). There was a reduction in the use of ceftriaxone, cefepime, daptomycin, and ertapenem. Importantly, an overall downwards trend of 26.6% in the use of parenteral management was evidenced [33].

Prospective audit and feedback strategies have also had positive impacts on mortality [34]. A study conducted in Brazil evaluated the impact of an AMS team intervention on 14-day and in-hospital mortality in patients with carbapenem-resistant enterobacteria bacteremia. There were 142 patients included in the study; 51 patients (35.9%) died within 14 days, of which 25.8% received an ASP team intervention vs. 44.7% without (*p* = 0.02), while 82 patients (57.7%) had in-hospital mortality, of which 52.2% were evaluated by an ASP team vs. 68.0% not evaluated (*p* = 0.08). The ASP team intervention was independently protective for 14-day mortality even after adjusting for septic shock status (*p* < 0.01) [34]. Finally, the ASP experience published by Pallares et al. in the four Colombian institutions included prospective audits and feedback for the antimicrobial guideline strategy [22]. The strategy played an important role in three of the four institutions. The multidisciplinary nature of the ASP team also included the nursing team, who positively contributed to the reduction in antimicrobial consumption; there was a statistically significant decrease in carbapenems, which had an important impact on local epidemiology [22].

### 2.3. Diagnostic Stewardship

The aim of interventions based on diagnostic stewardship is to improve the appropriate use and interpretation of tests to guide therapeutic decisions [13]. During this review, we identified a pre- and post-quasi-experimental single-center study conducted in Peru evaluating the impact of adding a rapid PCR-based blood culture identification (BCID) panel to the AMS protocol for patients with febrile neutropenia [35]. The results of this study demonstrated that adding this diagnostic tool to the AMS protocol improved the time to effective therapy from 10 h to 3.75 h compared to the pre-intervention period (*p* = 0.004). Interestingly, no outcome differences in mortality at 30 days, readmission, or relapse of bacteremia were evidenced during the study. Nevertheless, the length of stay was significantly reduced during the ASP intervention period. The authors highlighted the potential of introducing these diagnostic strategies to ASP and how this type of strategy might improve therapy success and direct cost savings [35].

### 2.4. Technological Tools: Electronic Clinical Decision Support Systems in AMS

For this category, there were only two studies found in LAC [36,37]. In Colombia, Ketcherside et al. [36] assessed the feasibility of extending commercial off-the-shelf software for infection control and ASP using three Colombian institutions. The software platform was delivered through a mobile application (app) and infection control and AMS features were added: clinical practice guidelines, hand hygiene (HH) documentation, and isolation precaution (IP) documentation. The software offered translation to almost any language and had flexible connectivity including an offline mode. The app also offered an interactive mode where users could track HH compliance or get support on therapeutic decisions for guideline management, among other interesting features. The results evidenced that the app was successfully implemented in all three participating institutions, albeit without the complete back-end integration of real-time patient data. HH and IP compliance tracking were the most used features among ASP staff, while treatment guidelines were most often used by physicians (reported as the most useful feature). In the end, it was a feasible tool that helped improve the time required for HH and IP decisions, physicians reported confidence in making antibiotic treatment decisions [36]. ASPs based on telemedicine were also proven to be effective for a remote community hospital in Brazil. In this study, infectious disease specialists from a high-complexity-level institution reviewed prescriptions from the community hospital uploaded to a secure website and received timely replies by e-mail or SMS text (median time to reply was 22 min); overall compliance with the recommendations of the infectious disease specialist was 100% (81/81 prescriptions) [37].

### 2.5. Pharmacy-Driven Protocols and Collaborative Practice Agreements (CPA)

Strategies including collaborative practice agreements between pharmacists and prescribers try to expand the role of pharmacists in patient care and broaden the role of pharmacists in ASP, positively impacting the AMS [13,14]. In LAC, we found some experiences of integrating pharmacists to ASP and different AMS strategies. Firstly, the study described the Brazilian experience of introducing a pharmacist to their prospective audit and feedback interventions for the ASP [32] and evaluated the differences in prescription and resistance trends before and after the integration of the pharmacist into the process. They concluded that the CPA between pharmacists and physicians in the ASP may have contributed to more rational prescription patterns of antimicrobials, cost savings, and changing bacterial resistant patterns [32]. A more recent prospective observational study conducted in a tertiary-level healthcare institution in Colombia described pharmaceutical interventions focused on broad-spectrum antibiotics in hospitalized patients [38]. Pharmacists trained in infectious diseases along with the infectious disease specialist modified antimicrobial therapy for a specific list of antibiotics and evaluated therapy, dosage, length of treatment, bacterial susceptibility, administration route of therapy, and adverse effects. Modifications were based on patients’ pharmacokinetic, pharmacodynamic and microbiologic conditions and comorbidities. With 258 patients included in the study, 16.1% were evaluated as receiving inappropriate therapy. Changes in prescription were implemented in 126 cases, with 82.5% of acceptance. The most common pharmaceutical interventions were the de-escalation of therapy and dose adjustment of antibiotics [38]. Finally, the study described in previous sections by Diaz-Madrid et al. [21] in Costa Rica applied multiple strategies (prospective audits and feedback, educational sessions, and development of guidelines), all led by a pharmacist-driven ASP, contributing to an overall reduction in antimicrobial use [21].

### 2.6. Economic Impact

Multiple studies have analyzed the impact of different AMS strategies on cost savings worldwide [16]. For the LAC region, there are different publications analyzing costs and the economic impact of implementing AMS strategies. Most of the publications focus on the impact of reducing antimicrobial consumption on hospital costs [24,30,32,39,40]. According to Fica et al. [39], the long-term effects of ASP and its impact on cost savings are not usually reported. An observational study comparing two long-term periods in a general hospital in Chile was designed, with a pre-AMS implementation (2005–2008) and post-ASP implementation (2009–2015) evaluation aimed to assess the effect the ASP had on expenditure, consumption, and AMR, as well as to estimate the contribution of competitive biddings on cost savings. The ASP implemented prospective audit and feedback interventions carried out by three infectious diseases specialists. The economic impact of the ASP from the pre-ASP evaluation to the last year evaluated (6-year period) was estimated in relation to antimicrobial consumption and competitive biddings. The results demonstrated that the ASP had a direct effect on cost savings, intensifying over time. There was an estimated 48.4% expenditure decrease in 2015, equivalent to cost savings of USD 190,000 [39]. A study presented previously by Magedanz et al. also reported a 69% reduction in hospital antibiotic costs after the implementation of an ASP [32]. In a study conducted by Rojas-Bonilla et al. [30] with ASM interventions based on antimicrobial restriction, they concluded that the reduction in costs was not significant for gentamicin, vancomycin, meropenem, cefotaxime, ceftazidime, or imipenem, with a non-significant increase in costs for amikacin, piperacillin/tazobactam, cefepime, and levofloxacin [30]. Other findings demonstrate that clinical and economic outcomes are significantly better for patients treated according to antibiotic flowcharts; flowcharts for common infections seem to be a very efficient way to implement prompt and appropriate empiric antibiotic therapy in some areas, like the emergency room [41]. Finally, another very interesting publication found was a formal economic analysis of two different AMS strategies implemented in a university hospital in Brazil [40]. This cost-effectiveness study developed a Markov model comparing the cost-effectiveness of two ASP modalities. A conventional (simplified stewardship program) and bundled (more comprehensive) ASP included prospective audits and feedback, face-to-face interventions to improve therapy, and guidelines for empiric therapy based on microbiological data from a microbiology lab. The model compared the two strategies with clinical data from a retrospective cohort using 30-day mortality as the main outcome. The bundled strategy proved to be more expensive, but more efficient; conducting a sensitivity analysis and cost-effectiveness acceptability curve, the study concluded that the bundled ASP strategy was effective, efficient, and affordable, making it more cost-effective than the other [40].

## 3. Expert Comments

The AMS experiences discussed above, as shown in Table 1, demonstrate the increased implementation of ASP in LAC. Addressing AMR is of the utmost importance as, according to the WHO, the consequences of inaction can be catastrophic, which is why national action plans have been called for by global health institutions, with AMS at the forefront [3]. This review illustrates a positive LAC trend in publishing AMS experiences, especially in recent years, which highlights the growing interest and sharing of interventions and their impact on hospital processes and healthcare outcomes. Regardless of finding mostly grey literature on this topic [12], there is clear progress in AMS in the region. There is still a lack of peer-reviewed publications on this topic in the majority of the countries in LAC, with most publications found in Brazil and Colombia [10,12]. The varied interventions implemented and the integration of multiple strategies within each hospital are important for the success of ASP, and there is a coherent tendency in the interventions analyzed [13,14]. The literature presented throughout this review illustrates that there is Latin American progress in the creation of specific (tailored) antimicrobial guidelines for particular healthcare settings. The studies found for antimicrobial preauthorization and restrictive measures showed contrasting results for antimicrobial consumption. Dolotario et al. in Brazil and Romo-Castillo et al. in Ecuador discussed the concept of ‘squeezing the balloon’ as a feature of restrictive measures, increasing antimicrobial consumption and resistance in some antibiotics as a collateral effect of the interventions in LAC [26,29,31]. In terms of prospective audits and feedback, the Brazilian study that implemented this strategy [33] shows a valuable perspective of AMS outside of high-complexity hospitals. The results of the published studies presented based on prospective audits and feedback demonstrated a positive impact. During this review, we were not able to find more than one study specifically based on diagnostic stewardship, maybe due to a lack of implementation of this useful strategy, which is identified as a highly valuable approach for AMS in LMICs [42], or the lack of publications for the LAC region. Regarding pharmacist–prescriber relationships and the benefits of pharmacist-driven protocols and CPA to improve AMS [21,32,38], the results emphasize the value of incorporating the pharmacy as an active member of the ASP. Most studies found applied multiple strategies, which highlights the importance of integrating interventions and different approaches for successful ASP implementation in healthcare settings. Quantitative tools have been adapted and developed in LAC in order to measure ASP development in healthcare institutions based on the LAC context in order to improve and intervene in AMS standards; the utility of using these tools as a starting point and evaluating progress are critically important to achieve the success of ASP, as presented by a recent study conducted by Pallares et al. [43]. The literature found for LAC demonstrates that there are diverse changes in cost savings based on the different AMS interventions implemented, but robust and systematic approaches to cost savings in healthcare settings related to the implementation of AMS strategies are difficult given the variability among healthcare systems, billing procedures, and, particularly, the lack of structured communication among healthcare staff and the administrative and financial decision-makers within institutions. The evidence presented for the economic impact of AMS has immense value for decision-makers in healthcare that can utilize these studies to support ASP and prioritize interventions for AMS specifically in LAC hospitals [24,30,32,39,40]. Social and cultural factors (such as collaboration between specialties and different levels of care and working conditions) have been shown to be important behavioral determinants in healthcare professionals for following ASP interventions and might represent barriers to the implementation and success of AMS strategies in the LAC region, as commented by Favre et al. [10]. The education and the shared discussion of AMS strategies for healthcare workers outside ASP teams can contribute greatly to overcoming some of these barriers and improving ASP development in LAC hospitals [13,25,43]. Unfortunately, AMS policy in Latin America is still missing in many countries. It is important for the region that each country’s government supports the creation of AMS policy and regulates the implementation of ASP programs by law, including investing in improving infrastructure. In turn, hospital leadership support, communication, and tracking and monitoring will improve within healthcare institutions. In addition to government support, education and training are needed to customize ASP programs for each hospital based on each country’s context and each institution’s ASP development. It is greatly acknowledged that the lack of (a) specific funding, (b) organizational structures supported by hospital leadership, and (c) clear communication of ASP activities and progress to leaders can hinder ASP development, as each of these is necessary for AMS success [43].

## 4. Call for the Future

Even though this review demonstrated progress in implementing ASPs in LAC countries, there is still a lot of work to do on this very important global health issue. There is a special need to strengthen leadership from the ‘top down’ and gain support through policy and financing [3,10]. Publications focusing on AMS strategies tailored to the pediatric community are still lacking and, even if there is some LAC evidence published on the topic, there is also room for improvement towards ASP efforts for this population [44]. It is also necessary to standardize outcome measures, which should be customized to each hospital’s ASP development and needs. However, hospitals and their ASPs are also struggling with identifying appropriate measures of success for such programs. Each hospital should define its goals and outcomes based on its own needs, as well as use validated tools that allow them to measure and compare AMS outcomes in the region and highlight opportunities for improvement. LAC has diverse sociopolitical environments, economies, and health systems [10]. However, there are some common important areas that should be emphasized for ASP success: increases in hospital budgets and resources to establish ASPs supported by the government, better investment in microbiology laboratories, the implementation of electronic health records (from which antibiotic use and other data can be obtained), incorporating pharmacists dedicated to AMS, enlisting financial decision-makers from institutions, and education to change behavioral determinants. There is a need in LAC to address the issues previously discussed, with investment from hospital administrators as well as the government to create higher ‘standards’ that can be implemented by healthcare institutions in the region.

## Figures and Tables

**Figure 1 antibiotics-12-01342-f001:**
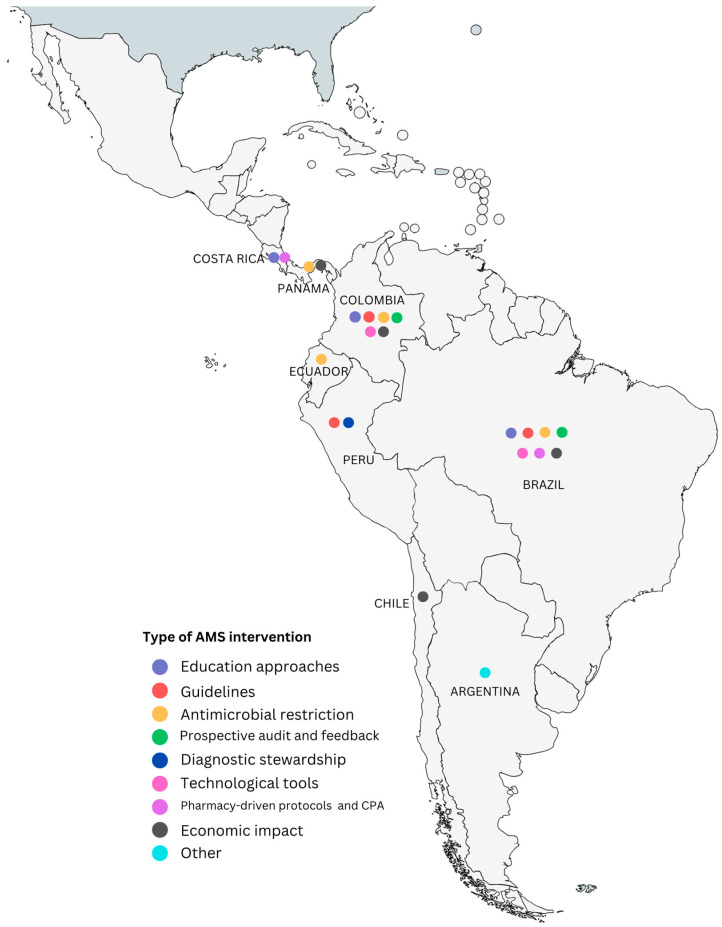
Latin American countries with published AMS interventions classified according to the common topics framework created. “Other” refers to publications about ASM in the country that could not be classified within the categories.

**Table 1 antibiotics-12-01342-t001:** Publications of AMS interventions in the LAC region.

Type of Intervention	Country	Publication Year	Reference
Education approaches	Brazil	2019	[23]
	Colombia	2012, 2020, 2022	[22,24,25]
	Costa Rica	2020	[21]
Guidelines	Brazil	2014	[27]
	Colombia	2012,2017,2022	[22,24,26]
	Peru	2019	[28]
Antimicrobial restriction	Brazil	2022	[31]
	Colombia	2017	[25]
	Ecuador	2022	[29]
	Panama	2020	[30]
Prospective audits and feedback	Brazil	2012, 2021, 2022	[32,33,34]
	Colombia	2022	[22]
Diagnostic stewardship	Peru	2023	[35]
Technological tools	Brazil	2013	[37]
	Colombia	2020	[36]
Pharmacy-driven protocols and collaborative practice agreements	Brazil	2012	[31]
	Colombia	2020	[38]
	Costa Rica	2020	[21]
Economic impact *	Brazil	2012, 2016	[32,40]
	Chile	2018	[39]
	Colombia	2012, 2017	[24,41]
	Panama	2020	[30]

Publications explaining ASP interventions in LAC healthcare settings and their impact are categorized by common and fundamental approaches [14]. * Economic impact was created as a separate category, even if studies contained different types of interventions.

## Data Availability

Not applicable.

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
