# Peer review of "Antimicrobial Stewardship Programs in Latin America and the Caribbean: A Story of Perseverance, Challenges, and Goals"

_antibiotics, 2023, doi:10.3390/antibiotics12081342_

Round 1
Reviewer 1 Report
The manuscript is well developed and written in good English. However, methodology section is very vague; how was the search performed, and how were the analyzed papers in the review scoped, assessed and finally included in the review?
The section on expert comments is not clear what it is based on; is it interviews with stakeholders, or drawn from different papers? THis should also be clearly described in the methodology section.
In the "Call for the future" section, the authors talk about behavioural determinants, which are not defined or described in the body of the manuscript (although the same are present in the analysis). Better link should be provided between the analysis and the recommendations for the future.
The manuscript adds to the body of knowledge in terms of providing good review of existing literature, however it needs revision on the points above.
Reviewer 2 Report
The authors have conducted a comprehensive review of the topic. The review is well planned and reported. Few suggestions from my side are as:
1. The authors should include search strategy in detail and flowchart for study selection (as mentioned in PRISMA guidelines).
2. Key characteristics of the publications on AMS interventions such as brief description of interventions including study design, settings, patient population, outcomes, strengths and limitations, implications etc. may be summarised in a separate table or incorporated in table 1.
Reviewer 3 Report
Reviewer comment
Manuscript ID:
The author should inform inclusion criteria before excluding any study.
page 2 line 87: Exclusions were done based on wrong publication type, wrong study population and AMS content.
To improve the flow of the discussion's topic, the author should present in the same order and terms.
page 2 lines 69-72 (Part 1): a) antimicrobial guideline implementation and monitoring b) diagnostic stewardship, 69 c) electronic clinical decision support systems in antimicrobial stewardship, d) pharmacy-driven protocols and collaborative practice agreements, and e) educational strategies [13, 14]. Another interesting topics, which we titled economic impact... vs pages 3-7 (Part 2)
The author should discuss the study's findings compared to Core Elements of Hospital Antibiotic Stewardship Programs from the Centers for Disease Control and Prevention (page 2 lines 55-60). The discussion of the ASP approach is general not specific LAC region. The author is supposed to discuss the barrier to those approaches' implementation in the LAC region (such as professional hierarchy, interprofessional collaboration, AMR national policy, and microbiology facilities). The author should discuss any determinant for the success of AMR in the LAC region.
none
Round 2
Reviewer 1 Report
I have no further comments on the revised manuscript.